# An Optimal Power Control Strategy for Grid-Following Inverters in a Synchronous Frame

**Juan F. Patarroyo-Montenegro** [1,*], **Jesus D. Vasquez-Plaza** [1], **Fabio Andrade** [1] **and Lingling Fan** [2]

[1] Sustainable Energy Center, University of Puerto Rico, Mayagüez, PR 00680, USA;
jesus.vasquez@upr.edu (J.D.V.-P.); fabio.andrade@upr.edu (F.A.)

[2] Electrical Engineering Department, University of South Florida, Tampa, FL 33620, USA; linglingfan@usf.edu

[*] Correspondence: juan.patarroyo@upr.edu; Tel.: +1-(787)363-2164

**Abstract:** This work proposes a power control strategy based on the linear quadratic regulator with optimal reference tracking (LQR-ORT) for a three-phase inverter-based generator (IBG) using an LCL filter. The use of an LQR-ORT controller increases robustness margins and reduces the quadratic value of the power error and control inputs during transient response. A model in a synchronous reference frame that integrates power sharing and voltage–current (V–I) dynamics is also proposed. This model allows for analyzing closed-loop eigenvalue location and robustness margins. The proposed controller was compared against a classical droop approach using proportional-resonant controllers for the inner loops. Mathematical analysis and hardware-in-the-loop (HIL) experiments under variations in the LCL filter components demonstrate fulfillment of robustness and performance bounds of the LQR-ORT controller. Experimental results demonstrate accuracy of the proposed model and the effectiveness of the LQR-ORT controller in improving transient response, robustness, and power decoupling.

**Keywords:** grid-following control; hardware-in-the-loop experiment; LQR controller; microgrids; optimal control; power control; three-phase inverters

## 1. Introduction

Due to the inherent low inertia present in voltage-source inverters (VSI), microgrid control is becoming an important object of study nowadays in order to improve penetration of renewable energy. Microgrid control is generally defined by levels. In [1], the authors present a hierarchical scheme that divides VSI controllers in different levels. First, voltage–current (V–I) control level is used to regulate the VSI current and voltage waveforms according to a specific reference without having significant harmonic distortion [2]. The power control level, or primary control level, regulates VSI power sharing to the microgrid. This power sharing could occur in grid-connected or islanded mode. For the grid-connected mode, the power grid imposes the voltage amplitude and frequency in the point of common coupling (PCC). In this mode, the VSI behaves as grid-follower. In islanded mode, the voltage and frequency in the PCC is determined by each of the generators in the microgrid. Typically, droop control is used in this control level because it allows to distribute power generation among VSI proportionally without communication. Higher control levels are in charge of maintaining microgrid's power quality and regulating power sharing from a microgrid to the power grid or even another microgrid.

Droop control is used with proportional-resonant (PR) or proportional-integral (PI) controllers to regulate power sharing as shown in Figure 1 [3,4]. To share power from the inverter to the main grid, the droop Equations (1) and (2) are used:

$$\omega \; = \; \omega_0 - m(P \, - \, P_0) \tag{1}$$

$$E \; = \; E_0 - n(Q \, - \, Q_0) \tag{2}$$

where $\omega_0$, $E_0$, $P_0$, and $Q_0$ are the operating frequency, amplitude, and reference values for the active/reactive power, respectively. Coefficients m and n are known as the droop gains. As analyzed in [2,5], the stability of the grid-connected IBG is mainly determined by the primary control level parameters. These parameters are the cut-off frequency of the power calculation low-pass filter and the coefficients m and n.

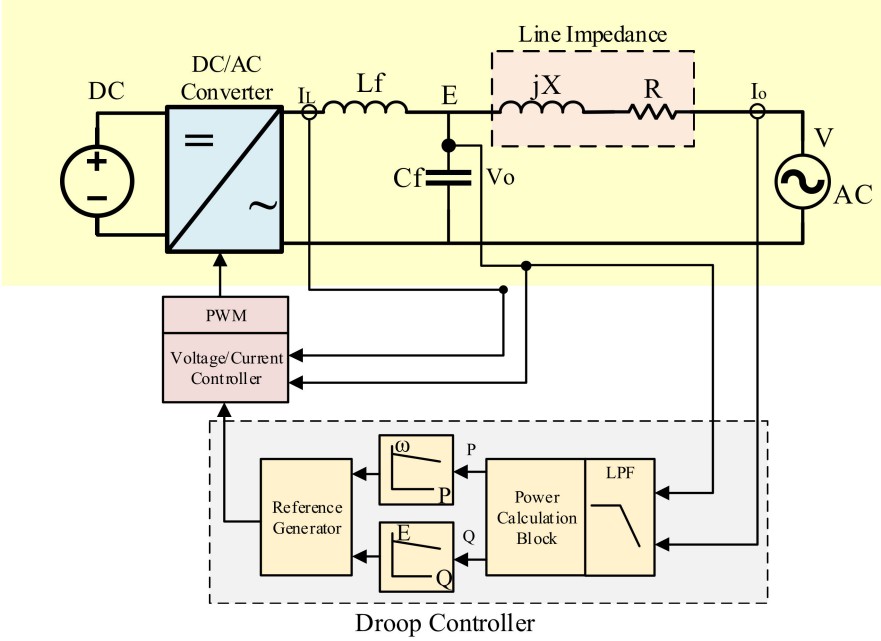

**Figure 1.** Typical droop control structure.

Due to the fact that droop controllers do not require a communication link between generators, this control method has become popular for VSI primary control. However, droop controllers are based on the assumption that active and reactive power are directly related to VSI output voltage amplitude and frequency. In addition, droop controllers rely on a low-pass filter that is used to separate the dynamics from the V–I and primary control levels. Thus, some common issues have been widely reported for droop controllers such as performance fulfilment [6–8], dependency on line impedance values [1,3,9], and frequency and voltage deviations from the nominal values [3].

Many approaches for improving droop control performance have been developed [2,10–17]. Most of them are based on adding supplementary loops to improve transient response, power quality, harmonic power sharing, or robustness. The gain selection of the supplementary loops is commonly based on heuristic methods and do not rely on an open-loop IBG model to use modern control design methods. Closing several loops of feedback control and designing with heuristic methods does not guarantee robustness of the closed-loop system [18].

For grid-following IBG control, it is common to implement output current controllers. Some of the most relevant publications in this area may be found in [19–28]. The open-loop state-space models for these controllers allow the use of modern control methods such as LQR, Kalman filters, or robust controllers based on $H_\infty$ theory. However, these models do not include power sharing dynamics and rely on a higher control level to regulate active and reactive power.

In this work, a state-space model that integrates V–I and power sharing dynamics for grid connected inverters is presented. This model is developed in a synchronous *d-q* frame to improve active and reactive power decoupling. The grid voltage is modeled as a constant vector that is included as an external known disturbance. In addition, the active and reactive power dynamics are included in the output equation of the state-space system. This type of model allows for using modern control analysis techniques such as singular value diagrams, Nyquist diagrams, root loci, and robustness margins estimations.

An optimal power sharing controller that minimizes a power tracking performance index is also presented. This tracking index measures the active and reactive power deviations and also measures the energy in the control inputs. The novelty of this work is based on how the nonlinear dynamics of the active and reactive power were linearized in the output of a state-space model in the *dq* frame. This allowed development a controller that minimizes the tracking error in the output of this state-space model. The methodology used in this work can be extended to other control methods such as *H∞* or model predictive control (MPC). Furthermore, this approach has many advantages over other approaches found in literature: First, this approach is intended to minimize the energy in the states and inputs, which means better transient response, reduced tracking error, and fewer power losses during transient responses. Second, the use of an LQR controller has many robustness properties regarding gain and phase margins as shown in [18]. Third, the controller is effective in reducing active and reactive power coupling. Finally, this approach does not use resonant filters that may affect sensitivity and robustness under parameter variations. The linear quadratic regulator with optimal reference tracking (LQR-ORT) controller was designed to meet certain robustness and performance margins under uncertainties in the LCL filter components. Hardware-in-the-loop (HIL) simulations and experimental results validate the results of this approach compared to a classical PR-Droop controller found in the literature. This work is an extension of [29] in the sense that it includes HIL simulations under component variations, experimental results in a physical setup, and deeper analysis and conclusions.

Section 2 shows the mathematical model in the *dq* frame that was used to compute the LQR-ORT controller. The LQR-ORT mathematical development is presented in Section 3. Section 4 presents the simulation and experimental results divided in three subsections: robustness and stability analysis, simulation results using HIL tools, and experimental results in a real testbench. Finally, the conclusion and future work are presented in Section 5.

## 2. Mathematical Model

The mathematical model is based on the circuit shown in Figure 2. The output of a three-phase inverter E is connected through an inductor–capacitor–inductor (*LCL*) output filter to a stiff voltage source V$_g$ that represents the main grid. Input inductor current, capacitor voltage, and output current are denoted by I$_l$, V$_c$, and I$_o$, respectively.

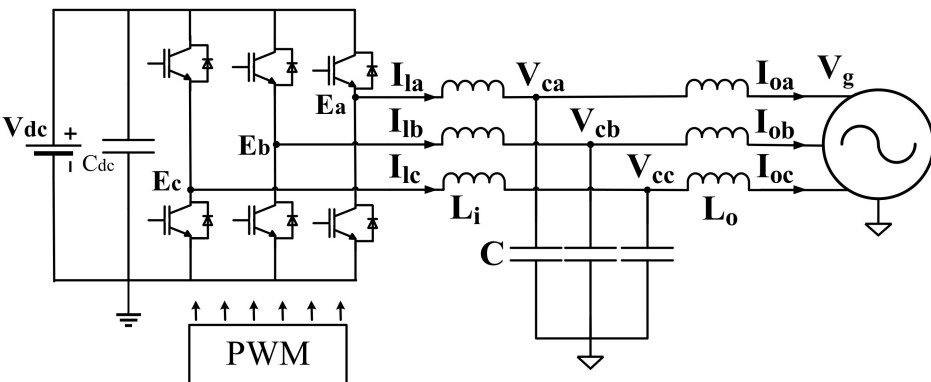

**Figure 2.** Three-phase inverter connected to the main grid.

The state-space model of this circuit for each phase in the *ABC* frame is given by

$$
\begin{bmatrix} \dot{V}_c \\ \dot{I}_l \\ \dot{I}_o \end{bmatrix} = \begin{bmatrix} 0 & 1/C & -1/C \\ -1/L_i & 0 & 0 \\ 1/L_o & 0 & 0 \end{bmatrix} \begin{bmatrix} V_c \\ I_l \\ I_o \end{bmatrix} + \begin{bmatrix} 0 \\ 1/L_i \\ 0 \end{bmatrix} E + \begin{bmatrix} 0 \\ 0 \\ -1/L_o \end{bmatrix} V_g
\tag{3}
$$

The main grid voltage $V_g$ is assumed as a balanced three-phase signal with an amplitude of $120V_{RMS}$ and an angular frequency $\omega_c = 2\pi 60Hz$. The state-space model (3) is transformed to a synchronous frame using the well-known *dq* transformation as follows [30]:

$$
w_{dq} = T_{dq} w_{abc}
\tag{4}
$$

where $W_{abc} = \begin{bmatrix} W_a & W_b & W_c \end{bmatrix}^T$ is some vector in the *ABC* frame representing each state variable and

$$
T_{dq} = \sqrt{\frac{2}{3}} \begin{bmatrix} \cos(\omega_c t) & \cos\left(\omega_c t - \frac{2\pi}{3}\right) & \cos\left(\omega_c t + \frac{2\pi}{3}\right) \\ -\sin(\omega_c t) & -\sin\left(\omega_c t - \frac{2\pi}{3}\right) & -\sin\left(\omega_c t + \frac{2\pi}{3}\right) \\ \frac{1}{\sqrt{2}} & \frac{1}{\sqrt{2}} & \frac{1}{\sqrt{2}} \end{bmatrix}.
\tag{5}
$$

The state-space model of the inverter in the *dq* frame is shown in (6). When $T_{dq}$ is synchronized with the main grid, $V_g$ becomes static with a constant value $V_{gdq} = \begin{bmatrix} \widetilde{V}_{gd} & 0 \end{bmatrix}^T$, where $\widetilde{V}_d$ represents the peak amplitude of the main grid voltage.

$$
\begin{bmatrix} \dot{V}_{cd} \\ \dot{V}_{cq} \\ \dot{I}_{ld} \\ \dot{I}_{lq} \\ \dot{I}_{od} \\ \dot{I}_{oq} \end{bmatrix} = \overbrace{\begin{bmatrix} 0 & \omega_c & 1/C & 0 & -1/C & 0 \\ -\omega_c & 0 & 0 & 1/C & 0 & -1/C \\ -1/L_i & 0 & 0 & \omega_c & 0 & 0 \\ 0 & -1/L_i & -\omega_c & 0 & 0 & 0 \\ 1/L_o & 0 & 0 & 0 & 0 & \omega_c \\ 0 & 1/L_o & 0 & 0 & -\omega_c & 0 \end{bmatrix}}^{A_{dq}} \overbrace{\begin{bmatrix} V_{cd} \\ V_{cq} \\ I_{ld} \\ I_{lq} \\ I_{od} \\ I_{oq} \end{bmatrix}}^{x_{dq}} + \overbrace{\begin{bmatrix} 0 & 0 \\ 0 & 0 \\ 1/L_i & 0 \\ 0 & 1/L_i \\ 0 & 0 \\ 0 & 0 \end{bmatrix}}^{B_{1dq}} \overbrace{\begin{bmatrix} E_d \\ E_q \end{bmatrix}}^{E_{dq}} + \overbrace{\begin{bmatrix} 0 & 0 \\ 0 & 0 \\ 0 & 0 \\ 0 & 0 \\ -1/L_o & 0 \\ 0 & -1/L_o \end{bmatrix}}^{B_{2dq}} \overbrace{\begin{bmatrix} V_{gd} \\ V_{gq} \end{bmatrix}}^{V_{gdq}}
\tag{6}
$$

The output of the state-space model is used to represent the active and reactive power received by the main grid as follows:

$$
y = \begin{bmatrix} P \\ Q \end{bmatrix} = \begin{bmatrix} V_{gd}I_{od} + V_{gq}I_{oq} \\ V_{gq}I_{od} - V_{gd}I_{oq} \end{bmatrix} = \frac{3}{2} \begin{bmatrix} \widetilde{V}_{gd} & 0 \\ 0 & -\widetilde{V}_{gd} \end{bmatrix} \begin{bmatrix} I_{od} \\ I_{oq} \end{bmatrix}.
\tag{7}
$$

Formulating the inverter model in the *dq* frame allows for expressing active and reactive power in the output equation using linear terms under nominal conditions in the main grid. This implies that not only can an optimal controller be computed to regulate power sharing but also modern robustness and stability assessment methods can be used to analyze power dynamics.

It is also important to remark that the system (6) must be discretized in order to implement it physically. Thus, the state vector must be augmented using a delay or integrator transfer function to account for the delay caused by the PWM switching [31]. The discrete-time integrator also allows for reducing the steady-state error in the closed-loop system. The resulting systems is given by

$$
X[k+1] = \begin{bmatrix} \overline{A}_{dq} & \overline{B}_{1dq} \\ 0_{2\times6} & I_{2\times2} \end{bmatrix} X[k] + \begin{bmatrix} 0_{6\times2} \\ T_s I_{2\times2} \end{bmatrix} E_{dq}[k] + \begin{bmatrix} \overline{B}_{2dq} \\ 0_{2\times2} \end{bmatrix} V_{gdq}[k]
\tag{8}
$$

where $X = \begin{bmatrix} x_{dq} & E_i \end{bmatrix}^T$, $T_s$ is the sampling period, and the upper bar $\blacksquare$ represents the discrete-time transformation. The auxiliary integral of the input is represented by $E_i$. The discrete-time state-space system can be rewritten as follows:

$$X[k+1] = \overline{A}_T X[k] + \overline{B}_{1T} E_{dq}[k] + \overline{B}_{2T} V_{gdq}[k]$$
$$Y = CX[k].$$

(9)

The output matrix C is given by

$$C = \frac{3}{2} \begin{bmatrix} 0 & 0 & 0 & 0 & \widetilde{V}_{gd} & 0 & 0 & 0 \\ 0 & 0 & 0 & 0 & 0 & -\widetilde{V}_{gd} & 0 & 0 \end{bmatrix}.$$

(10)

## 3. Discrete LQR-ORT Controller Design

The complete scheme of the LQR-ORT controller is presented in Figure 3. The LQR controller $K_d$ is obtained using a modified cost function that weights the tracking error and control input. The use of a modified cost function requires the computation of an auxiliary optimal reference tracking (ORT) matrix $K_v v$ to solve the optimization problem. Finally, since the classic LQR problem does not consider the influence of external disturbances, the superposition principle is used to account for the power contribution of $V_{dq}$. Superposition can be applied because the grid-following inverter was expressed in (6) as a linear state-space model that not only considers V–I dynamics but also include active and reactive power computations.

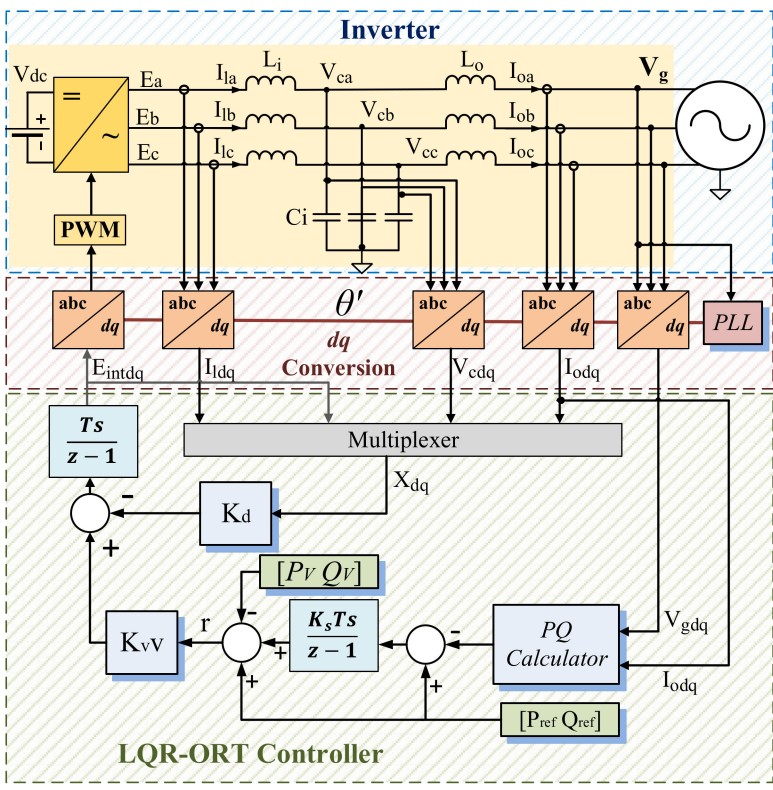

**Figure 3.** Control scheme for the LQR-ORT with known external input.

### 3.1. Computation of Suboptimal LQR Controller

Unlike classic LQR approaches, the LQR controller in this work must achieve a non-zero final state [18]. To achieve this, the power tracking error is defined by

$$e_{dq}[k] = CX[k] - r[k]$$

(11)

where r[k] is the reference for the LQR controller and $X[k]$ is the augmented state vector defined in (8). The discrete LQR-ORT cost function is given by [18]:

$$J[k_0] = e[T]^T S_p[T]e[T] + \sum_{k=k_0}^{T} \left\{ e[k]^T Q_p e[k] + E_{dq}^T[k]R_p E_{dq}[k] \right\} \tag{12}$$

where $S_p$ and $Q_p$ are both symmetric and positive semi-definite weighting matrices and $R_p$ is a symmetric positive definite weighting matrix. Note that unlike the classic LQR cost function, (12) includes only input and error signals and does not include the state vector $X[k]$. To solve this minimization problem, the modified discrete algebraic Riccati equation (DARE) must be solved [18]:

$$S_p = \overline{A}_T^T S_p \left( \overline{A}_T - \overline{B}_{1T} K_d \right) + C^T Q_p C. \tag{13}$$

The suboptimal controller matrix $K_d$ is given by

$$K_d = \left( \overline{B}_{1T}^T S_p \overline{B}_{1T} + R_p \right)^{-1} \overline{B}_{1T}^T S \overline{A}_T. \tag{14}$$

*3.2. Optimal Reference Tracking Matrix*

The reference signal r[k] introduces an additional equation to the optimization problem similar to (13). This equation is solved offline to obtain a control law that minimizes (12) [18]:

$$\nu[k+1] = \left( \overline{A}_T - \overline{B}_{1T} K_d \right)^T \nu[k] + C^T Q_p r[k] \tag{15}$$

where $\nu$ is an auxiliary matrix similar to $S_p$. Assuming a step reference signal r[k], and solving (15) with infinite horizon such that $\nu[k] = \nu[k+1]$:

$$\nu = \left[ I - \left( \overline{A}_T - \overline{B}_{1T} K_d \right)^T \right]^{-1} C^T Q_p. \tag{16}$$

For the LQR-ORT, the control law is given by [18]

$$E_{dq}[k] = \left( \overline{B}_{1T}^T S_p \overline{B}_{1T} + R_p \right)^{-1} \overline{B}_{1T}^T \left( -S \overline{A}_T X[k] + \nu r[k] \right), \tag{17}$$

or, in a compact form:

$$E_{dq} = -K_d X[k] + K_\nu \nu r[k]. \tag{18}$$

where $K_\nu = \left( \overline{B}_{1T}^T S_p \overline{B}_{1T} + R_p \right)^{-1} \overline{B}_{1T}^T$. The product $K_\nu \nu$ is the optimal reference tracking matrix (ORT). As shown in Figure 3, this matrix transforms a vector in the units of r[k] to a vector in the units of $E_{dq}[k]$ such that (12) is minimized.

*3.3. Main Grid Power Contribution Calculation Using Superposition Principle*

As $V_{gdq}$ is not considered in (12) to compute the LQR-ORT controller, its effects on the injected power must be subtracted from the raw reference signal $\begin{bmatrix} P_{ref} & Q_{ref} \end{bmatrix}^T$ to obtain the net power reference r[k] as shown in Figure 3. Thus, the superposition principle must be applied to the closed-loop system:

$$X[k+1] = \left( \overline{A}_T - \overline{B}_T K_d \right) X[k] + \overline{B}_{1T} K_\nu \nu r[k] + \overline{B}_{2T} V_{gdq}. \tag{19}$$

To analyze the power contribution of $V_{gdq}$, the closed-loop system (19) must be simulated with $r[k] = \begin{bmatrix} 0 & 0 \end{bmatrix}^T$. The power contribution of $V_{gdq}$ is computed using the expression

$$Y_V = \begin{bmatrix} P_V \\ Q_V \end{bmatrix} = \begin{bmatrix} V_{gd} & 0 \\ 0 & -V_{gd} \end{bmatrix} \begin{bmatrix} \bar{\bar{I}}_{od} \\ \bar{\bar{I}}_{oq} \end{bmatrix} \tag{20}$$

where $\bar{\bar{I}}_{od}$ and $\bar{\bar{I}}_{oq}$ are the output current values when (19) reaches steady state with nominal values of $V_{gdq}$. Since $Y_V$ and $K_v v$ are computed offline based on ideal components, there may be slight deviations in the experimental steady-state output values. To avoid these deviations, the integral of the power error is added to $r[k]$ as in Figure 3. This integral is calculated using a low-gain $K_s$ to not affect stability margins nor transient response. The net reference $R[z]$ is then defined as follows:

$$R(z) = \begin{bmatrix} P_{ref} \\ Q_{ref} \end{bmatrix} - \begin{bmatrix} P_V \\ Q_V \end{bmatrix} + \frac{K_s T_s}{z-1} \begin{bmatrix} P - P_{ref} \\ Q - Q_{ref} \end{bmatrix} \tag{21}$$

where $R(z) = Z\{r[k]\}$, with $Z\{\cdot\}$ being the *z*-transform operator. Terms P and Q represent the unfiltered measured power injected to the main grid.

## 4. Simulation and Experimental Results

For this section, the parameters shown in Table 1 were used. To perform the *dq* transformation and grid synchronization, a PLL-SOGI with parameters $K_{SOGI}$, $K_{pPLL}$, and $K_{iPLL}$ was implemented [32]. The state-space model (22) was obtained by substituting the parameters from Table 1 in (6). Then, the system was discretized and augmented with integrators using (8).

$$\begin{bmatrix} V_{cd} \\ V_{cq} \\ I_{ld} \\ I_{lq} \\ I_{od} \\ I_{oq} \\ E_{intd} \\ E_{intq} \end{bmatrix}_{[k+1]} = \begin{bmatrix} 0.432 & 0.016 & 9.085 & 0.342 & -9.113 & -0.343 & 0.283 & 0.006 \\ -0.016 & 0.432 & -0.342 & 9.085 & 0.343 & -9.113 & -0.006 & 0.283 \\ -0.044 & -0.001 & 0.711 & 0.026 & 0.283 & 0.011 & 0.049 & 0.008 \\ 0.001 & -0.044 & -0.026 & 0.711 & -0.011 & 0.283 & -0.008 & 0.049 \\ 0.044 & 0.001 & 0.283 & 0.011 & 0.7157 & 0.027 & 0.005 & 0.001 \\ -0.001 & 0.044 & -0.011 & 0.283 & -0.027 & 0.7157 & -0.001 & 0.005 \\ 0 & 0 & 0 & 0 & 0 & 0 & 1 & 0 \\ 0 & 0 & 0 & 0 & 0 & 0 & 0 & 1 \end{bmatrix} \begin{bmatrix} V_{cd} \\ V_{cq} \\ I_{ld} \\ I_{lq} \\ I_{od} \\ I_{oq} \\ E_{intd} \\ E_{intq} \end{bmatrix} + 0.0001 \begin{bmatrix} 0 & 0 \\ 0 & 0 \\ 0 & 0 \\ 0 & 0 \\ 0 & 0 \\ 0 & 0 \\ 1 & 0 \\ 0 & 1 \end{bmatrix} \begin{bmatrix} E_d \\ E_q \end{bmatrix} + \begin{bmatrix} 0.2837 & 0.007 \\ -0.007 & 0.2837 \\ -0.005 & -0.001 \\ 0.001 & -0.005 \\ -0.051 & -0.008 \\ 0.008 & -0.051 \\ 0 & 0 \\ 0 & 0 \end{bmatrix} \begin{bmatrix} V_{gd} \\ V_{gq} \end{bmatrix} \tag{22}$$

The values of the weighting matrices $Q_p$ and $R_p$ were selected so that the closed-loop system reaches steady state in less than 0.5s with an overshoot less than 10%. The following state feedback controller was obtained using (13) and (14):

$$K_d = \begin{bmatrix} -1154 & -58 & 6451 & 1193 & 22,624 & 2063 & 5158 & 70 \\ 58 & -1154 & -1193 & 6451 & -2063 & 22,624 & -70 & 5158 \end{bmatrix} \tag{23}$$

The desired refence was selected to be a constant vector $\begin{bmatrix} P_{ref} & Q_{ref} \end{bmatrix}^T$. Using (16), the optimal tracking matrix $K_v v$ was obtained:

$$K_v v = \begin{bmatrix} 117.9714 & 11.5883 \\ 11.5883 & -117.9714 \end{bmatrix}. \tag{24}$$

Outer integrator gain $K_s$ was selected to reduce steady-state error such that closed-loop dynamics and stability margins were not affected.

**Table 1.** Parameter Specification for the Linear Quadratic Regulator with Optimal Reference Tracking (LQR-ORT) Controller.

| Parameter | Symbol | Value |
|---|---|---|
| Grid Voltage | $V$ | 120 Vrms |
| DC bus Voltage | $V_{dc}$ | 350 V |
| Grid Frequency | $f$ | 60 Hz |
| Output Inductance | $L_o$ | 1.8 mH |
| Input Inductance | $L_i$ | 1.8 mH |
| Filter Capacitance | $C$ | 8.8 μF |
| Switching Frequency | $f_s$ | 10 kHz |
| Sampling Period | $T_s$ | 100 μs |
| Error Weighting Matrix | $Q_p$ | $5000 \times I_{2\times2}$ |
| Input Weighting Matrix | $R_p$ | $0.2 \times I_{2\times2}$ |
| Inner Integrator Gain | $K_i$ | 1 |
| Outer Integrator Gain | $K_s$ | 5 |
| SOGI gain | $K_{SOGI}$ | 0.7 |
| PLL Proportional Gain | $K_{pPLL}$ | 0.28307 |
| PLL Integral Gain | $K_{iPLL}$ | 7.5102 |

### 4.1. Robustness and Stability Analysis

The continuous LQR controller has guaranteed robustness properties such as infinity gain margin and a minimum phase margin of 60° [18]. However, these margins become reduced when the LQR controller is discretized. Studies performed in [33] and [34] show that minimum robustness margins for the discrete LQR are reduced depending on many factors such as eigenvalue location, controllability, and observability. To perform robustness and stability analysis on the LQR-ORT controller, the open-loop system $L(z) = K(z)G(z)$ shown in (25) was used.

$$L = K_d \left( zI - \overline{A}_{dq} \right)^{-1} \overline{B}_{1dq} \tag{25}$$

Stability margins were calculated using the disk margin method [35]. The disk margin method is used for estimating structured robustness under multiplicative uncertainties for MIMO systems. The disk-based margins are calculated considering all loop interactions and frequencies. Thus, the open-loop system L showed a gain margin of 12.03 dB and a phase margin of 52.23°, which is appropriate for inverter control applications.

From (19), it can be inferred that the stability of the power contribution of $V_{dq}$ depends on the eigenvalue locations of the closed-loop state matrix $\left( A_T - \overline{B}_T K_d \right)$. Thus, the stability of the complete closed-loop system is assessed by analyzing the return difference $(I + L)^{-1}$, which determines the closed-loop eigenvalues.

$$\lambda \left\{ (I + L)^{-1} \right\} = \lambda \left\{ \overline{A}_T - \overline{B}_T K_d \right\} \tag{26}$$

The operator $\lambda\{\cdot\}$ refers to the eigenvalue computation.

To assess robustness against parameter variations, the components in the *LCL* filter were defined as stochastic elements with a uniform variation of ±65% around parameters shown in Table 1. Thus, a Monte Carlo stability test was performed with 50 random instances of L based on parameter variations using the same controller $K_d$ [36]. This way, all interactions between parameter variations were evaluated instead of varying one parameter at the time. The continuous-time closed-loop eigenvalues of the nominal system and the system under component variations using the LQR-ORT controller are shown in Figure 4. The color bar indicates the maximum variation among each set of component values. Thus, the closed-loop eigenvalues tend to become unstable after a component deviation of −60%.

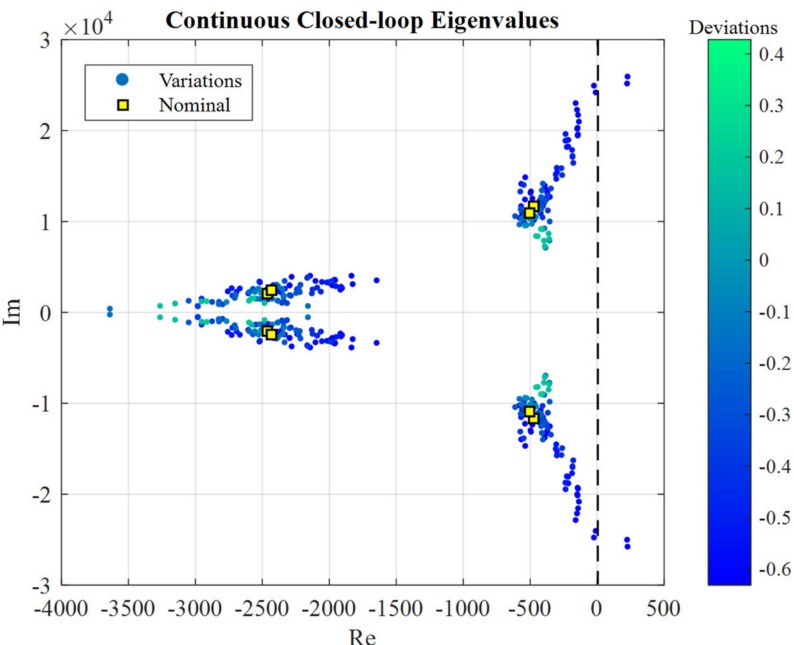

**Figure 4.** LQR-ORT discrete-time closed-loop eigenvalues of the nominal system and the system under variations using the same controller $K_d$.

### 4.2. HIL Experimental Robustness Assessment

To assess performance robustness, a real-time hardware-in-the-loop (HIL) experiment was performed. The LQR-ORT performance was compared against the classical proportional-resonant (PR) droop controller developed in [13]. The LQR-ORT controller and the PR-Droop controller were implemented in a dSPACE SCALEXIO real-time simulation platform.

The general scheme of the HIL experiment is shown in Figure 5. The circuit shown in Figure 2 was implemented in an OPAL-RT OP5700 real-time simulator using IGBT transistor models with a PWM switching frequency of 10 kHz. Each of the 50 component variations were programmed in the OP5700 in order to assess performance robustness under different component scenarios in a hardware-in-the-loop experiment (HIL).

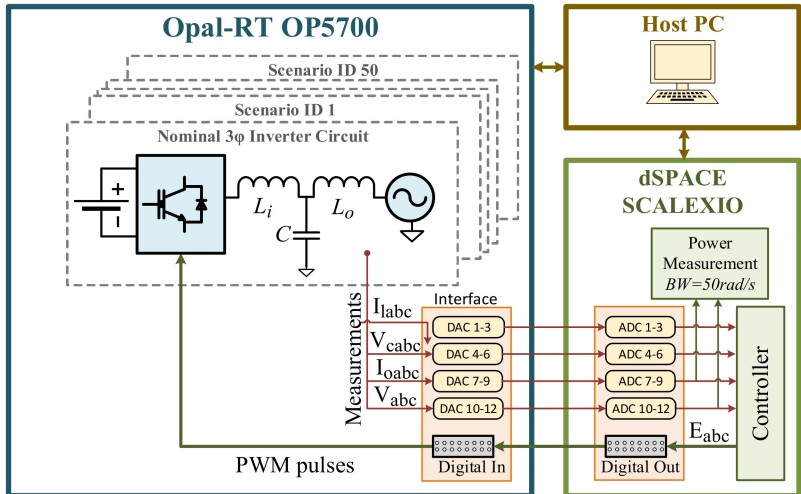

**Figure 5.** Real-time hardware-in-the-loop (HIL) experiment representation.

The dSPACE and the OP5700 were physically connected as shown in Figure 6. Step response robustness under component variations was assessed and summarized in Table 2. Each component

scenario ID is sorted from the most positive to the most negative deviation. Negative deviations occur when a component is under nominal values. Component variations between ±40% (ID 2 to ID 26) were stable for both controllers and their results are not included.

**Table 2.** HIL Experimental Robustness Assessment.

| ID | C (µF) | Li (mH) | Lo (mH) | Max Deviation | LQR-ORT Stable | PR-Droop Stable |
|---|---|---|---|---|---|---|
| Nom | 8.8 | 1.8 | 1.8 | 0% | Yes | Yes |
| 1 | 12.6 | 2.87 | 2.57 | 42.8% | Yes | Yes |
| ⋮ | ⋮ | ⋮ | ⋮ | ⋮ | ⋮ | ⋮ |
| 27 | 10.85 | 2.92 | 1.06 | −41.0% | Yes | Yes |
| 28 | 5.97 | 1.03 | 2.79 | −42.7% | Yes | **No** |
| 29 | 5.47 | 0.95 | 2.58 | −47.3% | **No** | **No** |
| 30 | 14.03 | 1.73 | 0.92 | −48.9% | Yes | Yes |
| 31 | 4.32 | 1.91 | 2.16 | −50.9% | Yes | Yes |
| 32 | 4.25 | 1.79 | 2.90 | −51.7% | Yes | Yes |
| 33 | 11.09 | 2.39 | 0.86 | −52.4% | Yes | **No** |
| 34 | 13.96 | 0.85 | 1.97 | −52.7% | Yes | **No** |
| 35 | 5.93 | 0.85 | 1.96 | −53.0% | **No** | **No** |
| 36 | 4.40 | 2.53 | 0.85 | −53.0% | Yes | Yes |
| 37 | 14.33 | 2.39 | 0.82 | −54.2% | **No** | **No** |
| 38 | 3.94 | 1.96 | 1.27 | −55.3% | Yes | **No** |
| 39 | 11.12 | 1.89 | 0.78 | −56.6% | Yes | Yes |
| 40 | 3.82 | 1.96 | 1.33 | −56.7% | Yes | **No** |
| 41 | 9.82 | 1.44 | 0.76 | −58.0% | Yes | Yes |
| 42 | 12.21 | 0.74 | 1.80 | −58.6% | Yes | **No** |
| 43 | 4.32 | 1.33 | 0.73 | −59.7% | **No** | **No** |
| 44 | 10.75 | 0.72 | 2.30 | −59.8% | Yes | **No** |
| 45 | 3.52 | 1.62 | 1.42 | −60.0% | Yes | **No** |
| 46 | 3.51 | 1.63 | 1.40 | −60.1% | **No** | **No** |
| 47 | 3.48 | 1.91 | 1.00 | −60.4% | Yes | **No** |
| 48 | 11.97 | 0.67 | 2.59 | −62.6% | Yes | **No** |
| 49 | 3.50 | 0.67 | 1.40 | −62.7% | **No** | **No** |
| 50 | 3.25 | 1.36 | 0.86 | −63.1% | **No** | **No** |

After a component variation of −42% (ID 28), the PR-Droop controller is more susceptible to becoming unstable than the LQR-ORT. Furthermore, the PR-Droop controller becomes completely unstable after a variation of 58.6% (ID 42), whereas the LQR-ORT becomes completely unstable after a variation of 62.7% (ID 49).

Figure 7 shows the response of both controllers to a pulsed reference of 300 W and 200 Var under nominal component values and scenarios 41 and 42. Each pulse had a width of 1.8 s and a time shift of 0.7 s to analyze decoupling between active and reactive power. To analyze transient response, power was measured using a first-order low-pass filter with a bandwidth of 50 rad/s as shown in Figure 6. The power measurement with the filter are outside the control loop and do not affect closed-loop dynamics. For nominal component values, the LQR-ORT controller shows a smooth transient response with low noise and low decoupling between active and reactive power. On the other hand, the PR-Droop controller shows oscillating responses with high switching noise and high coupling between active and reactive power.

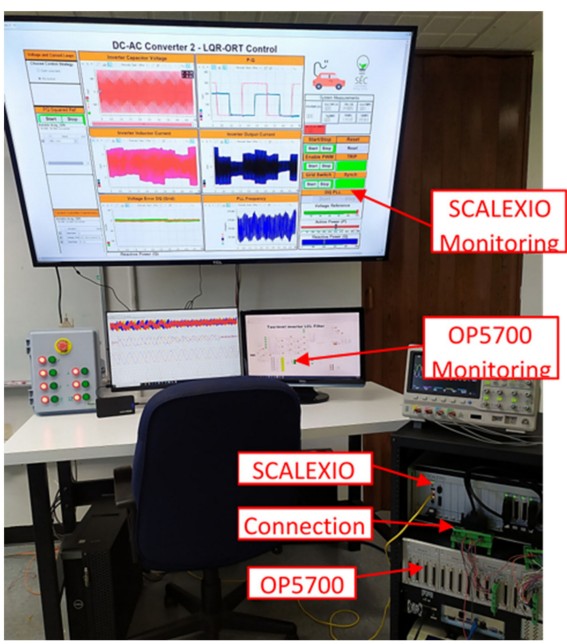

**Figure 6.** Photo of the HIL experimental setup.

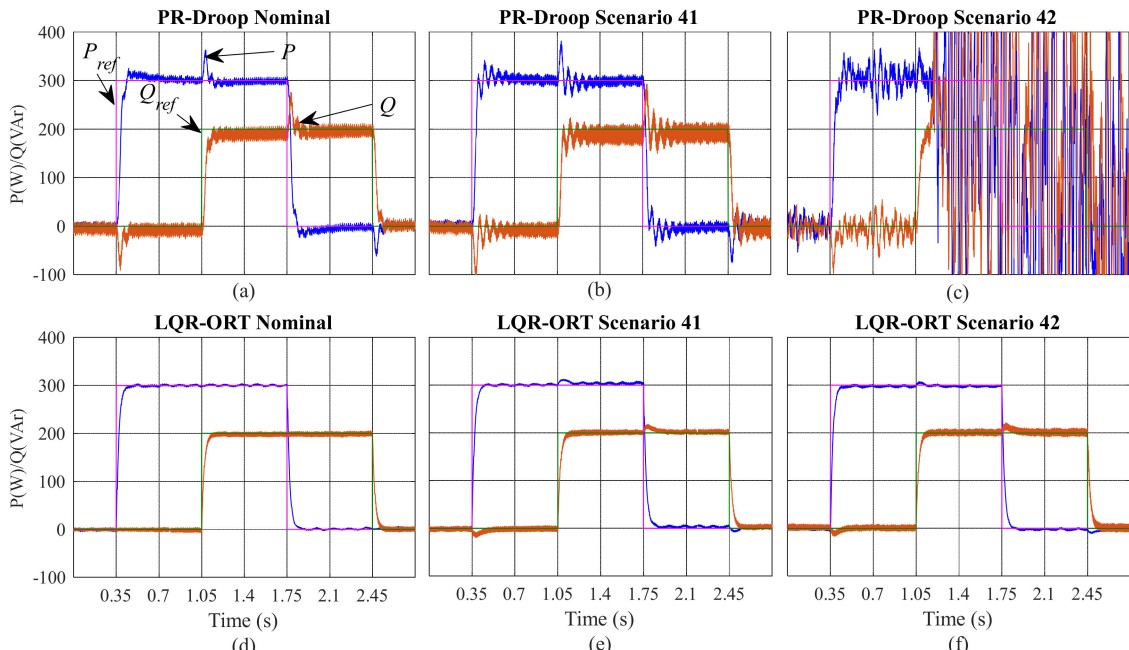

**Figure 7.** HIL experiment comparison between LQR-ORT and the PR-Droop controller for the nominal case and scenarios 41 and 42. (**a**) PR-Droop controller with nominal scenario. (**b**) PR-droop controller with scenario 41. (**c**) PR-Droop controller with scenario 42. (**d**) LQR-ORT controller with nominal scenario. (**e**) LQR-ORT controller with scenario 41. (**f**) LQR-ORT controller with scenario 42.

For scenario 41, The LQR-ORT transient response remains robust with some coupling between active and reactive power, whereas the PR-Droop response becomes more oscillating and the coupling increases. Finally, for scenario 42, the LQR-ORT shows a slight increase in switching noise and the PR-Droop becomes unstable. These results demonstrate that the LQR-ORT performance is more robust for component variation than the classic PR-Droop controller.

### 4.3. Experimental Results

Experimental results were obtained using the setup shown in Figure 8. This setup uses four Danfoss 2.2 kVA inverters with *LCL* filters and four sensor boxes with voltage and current LEM sensors. These four inverters may be dynamically connected to the AC bus. The AC bus may be also dynamically connected or disconnected from the main grid using solid-state relays handled with a PLC. To implement control algorithms, a dSPACE 1006 simulator was used to read signals from the sensor boxes and send the 10 kHz PWM pulses via optical fiber.

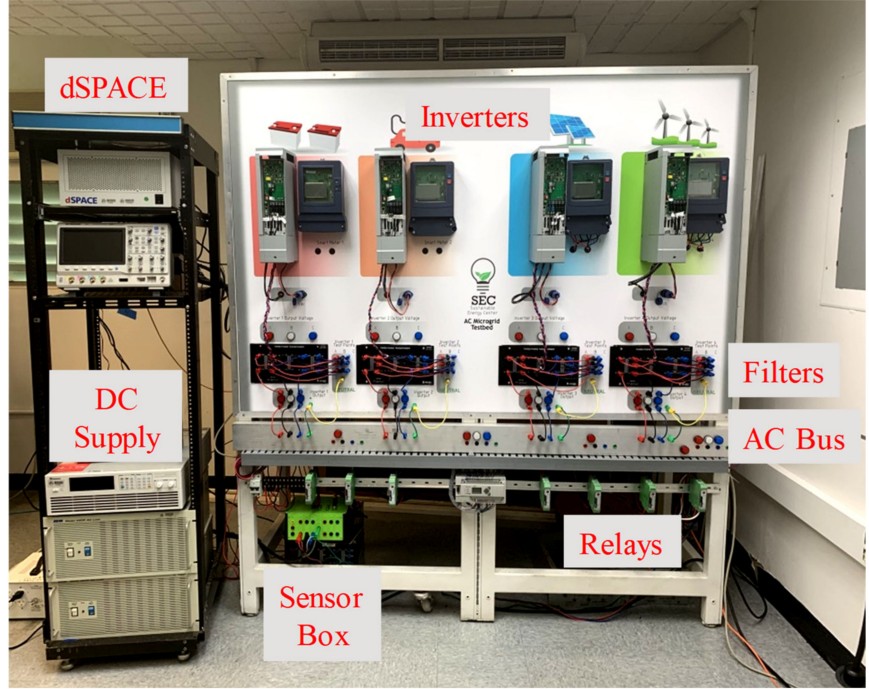

**Figure 8.** Photo of the experimental setup [37].

Experimental results for the LQR-ORT and the PR-Droop controller are shown in Figure 9. First, the inverter started without injecting energy to the main grid. A step reference of 300 W and 200 Var was set at $t_1 = 0.35$ s and $t_2 = 1.05$ s, respectively. The LQR-ORT shows lower settling time, less noise, and a smoother transient response under a step reference compared to the PR-Droop controller. In addition, the LQR-ORT controller shows a negligible steady-state error compared to the PR-Droop controller. In addition, at $t_1$ and $t_2$, the LQR-ORT controller shows better decoupling compared to the PR-Droop controller.

The decoupling of the LQR-ORT controller is produced by the *dq* transformation and the intrinsic robustness properties of the LQR controllers. In addition, the PR-Droop controller has less decoupling because droop $(\omega - V)$ curves are not completely independent [5]. The normalized cost was calculated using (12) divided by the highest achieved cost during the simulation. The normalized cost of the PR-Droop controller is about 2.92 times higher compared to the LQR-ORT controller. This implies that the LQR-ORT controller spends less energy tracking reference signals.

To evaluate the performance of the LQR-ORT controller under unbalanced and nonlinear loads, the scheme shown in Figure 10 was implemented. This scheme used a solid-state demultiplexer symbolized as "switch 2" that selected between linear balanced, linear unbalanced, and nonlinear loads. The linear load was selected to be an RL series circuit with R = 171.42 Ω and L = 0.4571 H. The linear unbalanced load was selected as an RL series circuit with $R_u$ = {171.42, 200, 600} Ω and L = 0.4571 H. Finally, for the unbalanced load, a three-phase noncontrolled rectifier with an RLC parallel circuit with $R_{NL}$ = 1200 Ω, $C_{NL}$ = 100 μF, and $L_{NL}$ = 2 mH was used.

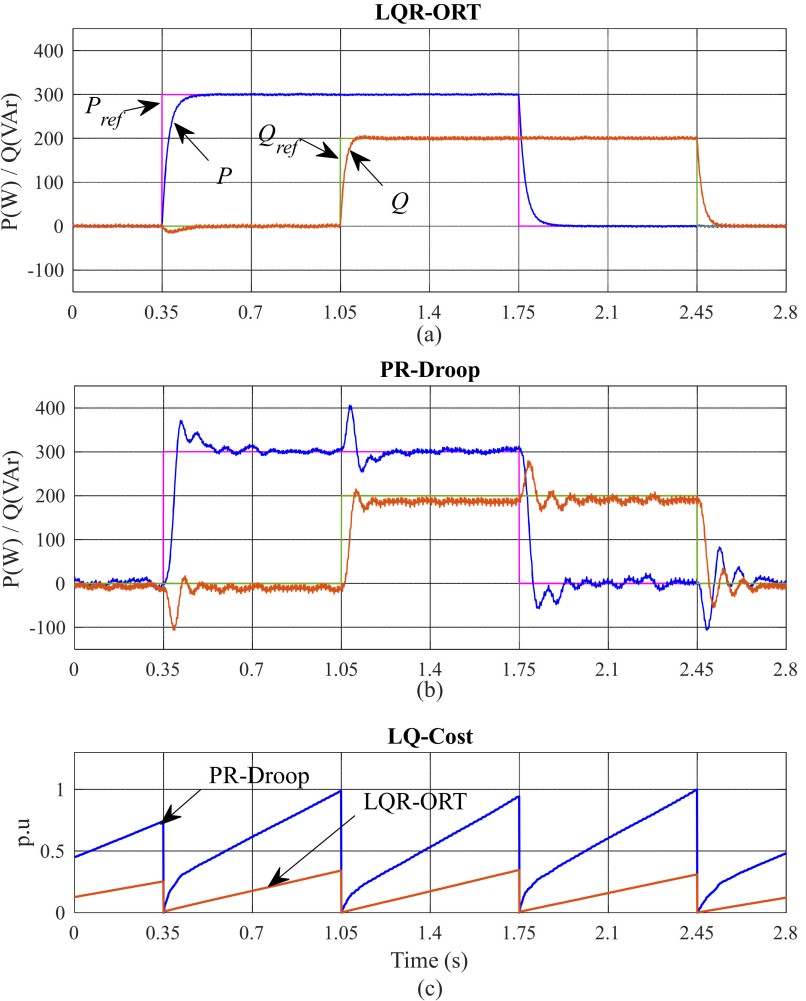

**Figure 9.** Experimental results comparing the LQR-ORT controller performance with the PR-Droop controller using the same reference signals. (**a**) LQR-ORT response. (**b**) PR-Droop response (**c**) LQ-Cost values.

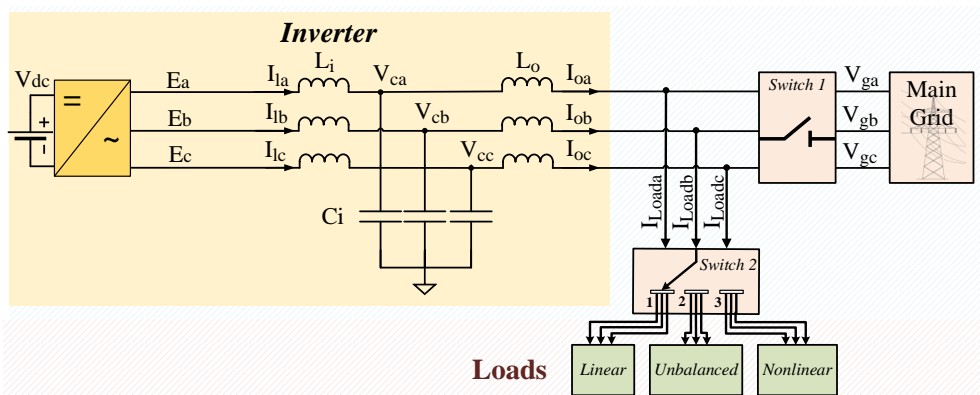

**Figure 10.** Experimental scheme for the evaluation of the performance of the LQR-ORT controller under unbalanced and nonlinear loads.

The results of this experiment are shown in Figure 11. First, the inverter injected 400 W and 400 VAr to the main grid with the linear balanced load connected to the output of the inverter (Figure 11a,b). At this time, the output current of the inverter had a Total Harmonic Distortion (THD) of 10.8% and the balanced linear load had a THD of 6.71%. At $t = 0.7$ s, the unbalanced load was connected.

This generated a small disturbance in the active power, which was restored in less than 0.2 s. In addition, the THD of the output current and the load decreased to 10.7% and 6.55%, respectively (Figure 11c). The low disturbance in the injected power and the decrease in the THD indicate that the LQR-ORT controller is robust under unbalanced loads. Finally, at *t* = 2.5 s, the nonlinear load was connected. This generated a slightly larger disturbance in the active power, which was restored in less than 0.2 s as well. In addition, the THD of the output current increased to 10.98% and the load current THD increased to 126.02%. (Figure 11d). It can be noticed that the THD of the output current did not have considerable increase under nonlinear load conditions. Despite the changes in the load conditions under nonlinear or unbalanced loads, the injected power of the LQR-ORT remains accurate without distortion.

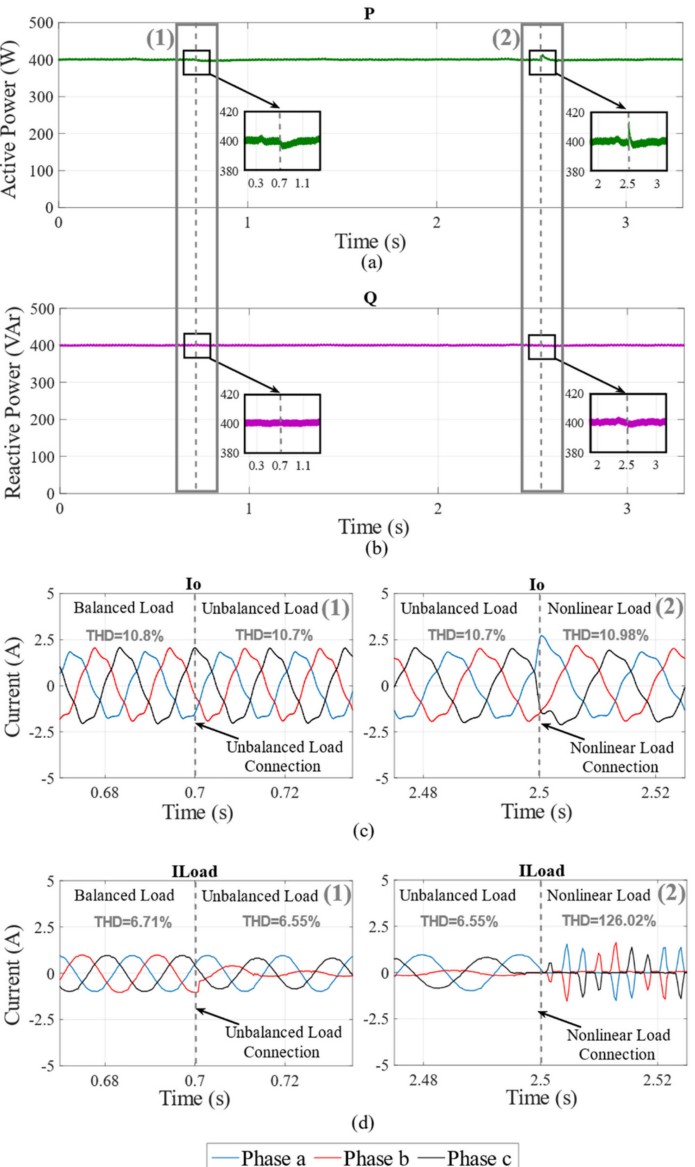

**Figure 11.** Experimental results evaluating the performance of the LQR-ORT controller under unbalanced and nonlinear loads. The inverter starts connected to a balanced linear load. At *t* = 0.7 s, event (1) occurs by connecting the unbalanced load. At *t* = 2.5 s, event (2) occurs by connecting the nonlinear load. (**a**) Active power. (**b**) Reactive power. (**c**) Output current showing the transition between balanced and unbalanced load (LEFT), and the transition between unbalanced and nonlinear load (RIGHT). (**d**) Load current showing the transition between balanced and unbalanced load (LEFT), and the transition between unbalanced and nonlinear load (RIGHT).

## 5. Conclusions

The LQR-ORT controller presented in this work yields a better transient response and higher decoupling compared to the classical PR-Droop control approach as can be seen in the experiment results. The model presented in this work integrates V–I and power sharing dynamics in a single state-space model. Using this new model allows performing modern analysis methods of stability and robustness. Stability and robustness analyses show that the LQR-ORT controller has robust performance and stability under uncertainties in components and multiplicative uncertainties. The robustness against component variations is important for heavy duty applications, where environmental conditions may affect component specification through time. It is also important to remark that, unlike the classic PR-Droop controller, the LQR-ORT controller does not require low-pass filters to perform power calculation. Thus, the power dynamics are not separated from voltage–current dynamics allowing to obtain smoother and faster transient responses.

**Author Contributions:** Conceptualization, J.F.P.-M., J.D.V.-P., F.A., and L.F.; Formal analysis, J.F.P.-M., L.F.; Investigation, J.D.V.-P.; Methodology, J.F.P.-M., J.D.V.-P., L.F. and F.A.; Project administration, F.A.; Resources, F.A.; Validation, J.D.V.-P.; Writing—original draft, J.F.P.-M. and J.D.V.-P.; Writing—review & editing, J.F.P-M and J.D.V.-P. All authors have read and agreed to the published version of the manuscript.

**Funding:** This research was funded by the United States Department of Energy, grant number DE-SC0020281.

**Conflicts of Interest:** The authors declare no conflicts of interest.

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
