# Peer review of "An Optimal Power Control Strategy for Grid-Following Inverters in a Synchronous Frame"

_applsci, doi:10.3390/app10196730_

Round 1
Reviewer 1 Report
This work presents a power control strategy based on the Linear Quadratic Regulator with optimal reference tracking (LQR-ORT) for a three-phase inverter-based generator using LCL filter.
1-Authors extended their previous work presented in IEEE conferences, i.e., references [29] and [37], by providing more simulation/experimental results. However, since the mathematical model, the designed controller, and the experimental setup have been already appeared in the authors' previous publications. Therefore, the contribution of this work is not significant enough to be published again.
2-The provided experimental results are not sufficient. For example the current harmonics and total harmonics distortion have not been provided.
3- Authors did not evaluate the performance of the utilized controller under unbalanced and nonlinear load conditions. Experimental results for unbalanced conditions should be provided.
Author Response
Reviewer Comments and Answers
Dear Reviewer,
Thank you for your comments. Your insights on this manuscript is of high value to improve the quality of our work. Following, you may find the answers to your requests.
This work presents a power control strategy based on the Linear Quadratic Regulator with optimal reference tracking (LQR-ORT) for a three-phase inverter-based generator using LCL filter.
1-Authors extended their previous work presented in IEEE conferences, i.e., references [29] and [37], by providing more simulation/experimental results. However, since the mathematical model, the designed controller, and the experimental setup have been already appeared in the authors' previous publications. Therefore, the contribution of this work is not significant enough to be published again.
Thank you for your comment. This is an important matter to discuss. The concept of the work reported in [29] was simulated in 2019, when this research started. This concept worked well because the simulation was made under ideal conditions. Also, this article was submitted to a conference meeting which had reduced space to report mathematical development and results (5 pages). This work lacked proper stability and robustness analysis, which is highly required for journal articles. The development of the robustness analysis, the experimental results, and the Hardware-in-the-loop tests were considered an important addition to our work since they demonstrate the applicability of it. In addition, we consider that the Hardware-in-the-loop experiment, where we change between multiple scenarios with multiple output filter parameters, is of high interest for the scientific community. Finally, in [37] we did not report the HIL setup since we developed it for this publication.
2-The provided experimental results are not sufficient. For example the current harmonics and total harmonics distortion have not been provided. Authors did not evaluate the performance of the utilized controller under unbalanced and nonlinear load conditions. Experimental results for unbalanced conditions should be provided.
Thank you for your comment. We performed an additional experiment to show the controller performance under unbalanced and nonlinear loads. In addition, we analyzed the Total Harmonic Distortion (THD) of the output current to determine if it was affected by different types of loads at the output. You may find the addition of these experiments and the analysis on page 13, lines. 262 to 281.

Reviewer 2 Report
Review of the article "An Optimal Power Control Strategy for Grid-Following Inverters in a Synchronous Frame"
The article was written very carefully, in good scientific language and methodically correct.
A very carefully conducted line of thought allows you to follow the scientific path. In particular, a very good mathematical model.
As a supplement, it would be worth mentioning what was the novelty proposed by the Authors and what was the scientific purpose of the deliberations?
I highly appreciate the content and diligence.
Author Response
Dear Reviewer,
Thank you for your response. It means a lot to receive such comments from the scientific community. We addressed your request regarding to emphasize the novelty of our work in lines 77-80.
If you have further questions, we will gladly address them.
Reviewer 3 Report
The paper is a very interesting approach to develop an optimal power control strategy for grid-2 following inverters in a synchronous frame. The optimal reference tracking controller designed in the paper benefits from (LQR-ORT) and uses a three-phase inverter-based generator (IBG) using LCL filter. The paper is written very well and the results presented are very promissing. Good use of graphics in the paper which makes it a perfect candidate for publication. The presented approach is also implemented on areal-time system which makes it even more interesting. My only concern is some of the equations need to be presented in a better format. In some matrixes, numerical values jump out of the brackets which may be improved.
Author Response
Dear Reviewer,
Thank you for your response. It means a lot to receive such comments from the scientific community.
Regarding your comments on the equations, we used the journal's template in MS word. We agree with your concern and hope that in the publication process with the editors we could improve the format in the equations. We tried to increase the size of the brackets and we were unable to do it.
If you have further questions, we will gladly address them.
Round 2
Reviewer 1 Report
I would like to thank authors for their time and efforts addressing my comments/suggestion. Authors explained that they did not described the HIL setup or did not provide comprehensive mathematical explanation due to the page limit of their previous publications. Authors also added some results concerning unbalanced and nonlinear loads, which are necessary. However, formatting and size of the added curves/results are not consistent with other presented results. Also, these figures have no or incomplete captions. Authors are required to provide a clear and consistent revised manuscript rather than incomplete or rushed revised version.
In general, it is not clear how this work will contribute to the field as it has already been published, but, I leave this to Associate's Editor for final decision without me reviewing it again.
Author Response
Dear Reviewer,
Thank you for your comments. We modified Fig. 11 according to your suggestions. First, we separated the figure into 4 subfigures: (a), (b), (c), and (d). Figures (a) and (b) are used to show the effects of changing the loads on the output power. Figures (b) and (c) are used to show the current harmonic distortion and waveforms under changes in loads. Also, we added tags to some parts of the figure to clarify the information to the reader. Finally, we modified the figure caption to have a better explanation of the figure content.
We hope that with these improvements you find the figure adequate. If you consider that something else should be done, we will gladly address your suggestions.